# Pulsation Reduction Using Dual Sidewall-Driven Micropumps

**DOI:** 10.3390/mi14010019

**Published:** 2022-12-21

**Authors:** Takuto Atsumi, Toshio Takayama, Makoto Kaneko

**Affiliations:** 1Department of Mechanical Engineering, Tokyo Institute of Technology, 2-12-1 Ookayama, Meguro-ku, Tokyo 152-8552, Japan; 2Graduate School of Science and Engineering, Meijo University, 1-501 Shiogamaguchi, Tempaku-ku, Nagoya 468-8502, Japan

**Keywords:** microfluidic device, micropump, sidewall-driven, pulsation, peristaltic

## Abstract

Single-cell manipulation in microfluidic channels at the micrometer scale has recently become common. However, the current mainstream method using a syringe pump and a piezoelectric actuator is not suitable for long-term experiments. Some methods incorporate a pump mechanism into a microfluidic channel, but they are not suitable for mass production owing to their complex structures. Here, we propose a sidewall-driven micropump integrated into a microfluidic device as well as a method for reducing the pulsation of flow. This sidewall-driven micropump consists of small chambers lined up on both sides along the main flow path, with a wall separating the flow path and each chamber being deformed by air pressure. The chambers are pressurized to make the peristaltic motion of the wall possible, which generates flow in the main flow path. This pump can be created in a single layer, which allows a simplified structure to be achieved, although pulsation can occur when the pump is used alone. We created two types of chips with two micropumps placed in the flow path and attempted to reduce pulsation by driving them in different phases. The proposed dually driven micropump reduced pulsation when compared with the single pump. This device enables precise particle control and is expected to contribute to less costly and easier cell manipulation experiments.

## 1. Introduction

Microfluidic devices are used to manipulate and observe single cells and have been applied in various pharmacological and cytological studies [1,2,3,4,5]. For example, Myrand-Lapierre et al. [6] developed a chip to examine red blood cell deformability, whereby multiple red blood cells are passed through a constricted channel using a microfluidic device in parallel. In addition, Sakuma et al. [7] evaluated erythrocyte fatigue by creating a narrow throat in a microfluidic channel and subjecting erythrocytes to repeated back-and-forth motion along the narrow throat. This narrow throat mimicked the capillaries, and the experiment reproduced the environment in which red blood cells circulate throughout the body. These studies contributed to the understanding of the relationship between the deformability of red blood cells and various diseases.

In these experiments, the position of the cells or particles was controlled by pushing and pulling all liquid in the channel. The most common method for precise position control is to connect the syringe pump directly to the microfluidic channel and operate the syringe with a piezoelectric actuator. However, piezoelectric actuators typically have a short stroke length of several tens of micrometers and cannot be operated for long periods, as they saturate after a certain point, as shown in Figure 1a. Mizoue et al. [8] reported that a piezoelectric actuator reaches its maximum length in less than 10 min. In some cell position manipulation experiments, the duration of the experiment can be 30 min or longer, and the piezoelectric actuator must be reset each time it stops, increasing the necessary time and effort. As a result, experiments cannot be performed continuously over a long period without pausing, which limits the number of experiments that can be conducted. Many companies offer external pumps that can be applied to microfluidic channels as an alternative to conventional methods. For example, a microfluidic pump developed by Blacktrace is capable of pulseless pumping and can be used in applications such as droplet generation [9]. However, these commercially available pumps are designed for continuous flow and are not suitable for applications that require fast response, such as position control. In addition, contamination must be avoided when conducting cell experiments. To avoid mixing the liquid in the pump with that in the channel by connecting the pump directly to the channel, the pump must be integrated into the channel device.

Currently, pumps that do not use piezo actuators or syringes are being studied, such as electroosmotic pumps [10,11,12,13]. Glawdel et al. [14,15] used electroosmotic pumps for cell culture by creating a chip that separated the high-electric-field area from the area where cells were cultured. Ye et al. [16] developed a high-flow rate electroosmotic pump in which the electrode did not come into contact with the sample by constructing a chip with a multi-layer structure that could separate the flow path and electrode layers. Although the flow rate can be easily regulated with the voltage, the ions generated via electrolysis are problematic for cell manipulation. Sealing the electrode and flow path with a membrane or gel can prevent or delay the inflow of ions; however, this further complicates the chip structure. Another example of a non-mechanical pump is the electrohydrodynamic pump [17,18,19,20]. This method uses electrohydrodynamic phenomena created by applying a voltage to the flow path to move the fluid. Unlike in the electroosmotic pumps described above, dielectric fluids are generally used as the working fluid. Electrohydrodynamic pumps are used in various applications, such as micromixers [21] and droplet [22] generation. However, as with electroosmosis pumps, the separation of the pump from the cell and the complexity of the structure limit cell manipulation.

Microfluidic devices are typically composed of soft and highly elastic polydimethylsiloxane (PDMS), and pumps have subsequently been developed to exploit their deformability. Xia et al. [23] fabricated a diaphragm micropump using an electrostrictive poly(vinylidene fluoride-trifluoroethylene)-based polymer as the actuator material. Zhao et al. [24] performed droplet generation experiments using a diaphragm pump integrated into the flow channel. Diaphragm pumps [25,26,27,28] are used in the micro total analysis system field; however, because of their structure, a single pump can only pump in one direction. In addition, the production of these pumps is complicated, considering their multi-layer structure, making them labor intensive and costly to construct.

Alternatively, a micropump can deform the channel wall [29,30,31,32]. Unger et al. [33] fabricated a chip with three driving channels in the upper layer of the main channel. By pressurizing in turn the driving channels with air pressure, the upper walls of the main channel became peristaltic, and flow could thus be generated. In addition, Suzuki et al. [34] developed a pump with nine piezoelectric actuators placed along a channel on its top surface. By deforming these piezoelectric actuators in sequence, the resultant traveling wave generated a flow. Yamamoto et al. [35] developed a pump that used a large, deformed polymer film as the bottom wall of a microchannel and applied vibrations to the film using piezoelectric actuators to generate traveling waves from the bottom wall of the channel. Various pumps have been developed to generate peristaltic motion or traveling waves by deforming the channel wall mainly by deforming the upper or bottom wall. These pumps are created by stacking multiple layers, which imposes a manufacturing burden because of their multi-layer structure, as shown in Figure 1b. Particularly, this manufacturing burden is an important problem considering that microfluidic channels are often discarded during cell experiments. In addition, these methods exhibit fluid pulsation in accordance with the driving cycle. This is because the deformed wall draws the surrounding fluid as it returns to its original shape. Pump pulsation is a limitation not only in the microfluidics field but also in macroscopic pumps and is used in positive-displacement reciprocating pumps, such as piston and diaphragm pumps. Nishikata et al. [36] applied a parallel-pump arrangement to a micropump. Three braille actuators were placed on the channel surface to create a peristaltic pump, with two of these pumps being placed in parallel to reduce pulsation. In addition, the displacement of each actuator was optimized to reduce pulsation through the analog control of the displacement of the braille actuators. In their study, by making the channel cross-section tunnel-shaped, it was completely cut off when pressurized by the actuators. Although completely blocking the channel can reduce the backflow, further studies are required to create a channel with a tunnel-shaped cross section.

In this study, we developed a sidewall-driven micropump that generates peristaltic motion by deforming the channel sidewalls and proposed a method using this approach to reduce pulsation, as shown in Figure 1c. The sidewall-driven micropump consists of a flow channel and a drive chamber within the same layer, thus allowing a microfluidic chip to be constructed in a single layer. The method of deforming the channel sidewalls cannot completely close the channel, regardless of the level of applied pressure, resulting in a large backflow when the pump is driven. To overcome this limitation, two micropumps were placed in the channel and driven in different phases to reduce pulsation, and two pump configurations were tested. In addition to the typical parallel arrangement, a series arrangement with pumps at both ends of the linear flow path was also evaluated. Overall, pulsation was reduced using the dual sidewall-driven micropump.

The remainder of this paper is organized as follows: In Section 2, we explain the principle of the sidewall-driven micropump and pulsation reduction method. In Section 3, we present the experimental results. In Section 4, based on the experimental results, we discuss the characteristics of the dual sidewall-driven micropump, and in Section 5, we provide concluding remarks.

## 2. Materials and Methods

### 2.1. Principle and Channel Design

As mentioned in the Introduction section, fluid transport by peristaltic motion is widely used and can be generated by applying displacements in turn using three or more actuators. The proposed sidewall-driven micropump is shown in Figure 2; the structure consists of a channel through which a suspension of cells or particles flows, with four chambers being placed on each side of the channel. The tubes are connected to these chambers, and air pressure can be applied to the chambers through the tubes. When air pressure is applied, the walls separating the chambers from the channels are deformed, pressurizing four pairs of chambers to induce peristaltic motion, thereby pumping fluid in the channel. However, the peristaltic pump generates pulsations in accordance with the pump drive cycle. Steps (1) to (4) in Figure 3a show the forward and reverse flows at each pumping step. In steps (1), (2), and (3), the fluid pushed away by wall deformation flows to the right. In step (4), when the second chamber from the right depressurizes and the surrounding fluid is drawn in, weak backflow occurs because the rightmost wall cannot completely block the flow path, as shown in Figure 3b. Considering that the upper and lower ends of the deformed wall are fixed, the channel cannot be completely blocked, regardless of the pressure. When the right end of the wall is opened in step (1), a larger backflow than that observed in step (4) occurs. If the drive can completely close the channel, fluid backflow can be reduced. However, the proposed sidewall-driven method cannot completely close the channel, and large pulsation is generated as a result.

To cope with pump pulsation, we propose a chip with two sidewall-driven micropumps installed in the flow path, driven in different phases to cancel the pulsation. Two flow paths are suggested for the pump arrangement, which are referred to as parallel and series pumps. The parallel pump, shown in Figure 4a, has a structure in which the flow paths merge downstream of each pump. The particle position manipulation in this study was assumed to occur at the point where the flow paths merge. The series pump shown in Figure 4b consists of pumps at both ends of a straight flow path. Particle positioning was assumed to be performed at the center of the channel, equidistant from the two pumps.

### 2.2. Microchip Fabrication

A silicon wafer was spin-coated with SU8-3050 (KAYAKU Advanced Materials, Westborough, MA, USA) at a target height of 100 µm and pre-baked at 95 °C for 45 min. SU8 was ultraviolet (UV)-irradiated using a maskless exposure system (PALET; NEOARK Corporation, Tokyo, Japan) in the same pattern as the channel geometry. After post-baking at 95 °C for 5 min, the mold was made thinner. PDMS (SILPOT184; Dow, Midland, MI, USA, base:curing agent = 9:1 (mass ratio)) was then poured into the mold for curing and to transfer the pattern. PDMS with the transferred flow path pattern was bonded to a glass slide with plasma treatment to create a flow path. Chips for the parallel and series pumps were also prepared as described above.

### 2.3. Experimental Setup and Procedures

The experimental setup is shown in Figure 5. Air was supplied from a compressor (AK-T20R; Max Co., Tokyo, Japan) to the solenoid valve via a regulator, and air pressure was applied from the valve to the drive chamber via a tube. The solenoid valve was controlled with a PC using a USB-connected digital input/output device (DO-16TY-USB; CONTEC Co., Osaka, Japan). The drive chamber was filled with pure water to prevent air bubbles from penetrating the channel walls. A suspension of 4.5 µm microbeads (Polybead Polystyrene 4.5 Microspheres; Polysciences, Philadelphia, PA, USA) flowed through the channel, and the movement of the microbeads was magnified using a microscope (IX73P1F; OLYMPUS, Tokyo, Japan) during pumping and captured using a high-speed camera (CHU130EX; SHODENSHA, Tokyo, Japan). The frame rate of this camera was 200 frames per second, with a shutter speed of 1/1000 s.

The experiments were carried out using single-pump, parallel-pump, and series-pump tips. The single-pump experiment was performed using only one of the series pumps. In parallel- and series-pump experiments, the two pumps were driven in five different phases: 0, π/4, π/2, 3π/4, and π. Bead movement was tracked in the observation area indicated by the rectangles in Figure 4a,b, whereas the forward and backward distances were measured using the pixel values. Air pressure was maintained at 0.20 MPa, and the pump was driven at 20 Hz in the order shown in Figure 3a. These conditions were experimentally determined to maximize the pump flow velocity.

## 3. Results

### 3.1. Single Pump

The experimental results are shown in Figure 6, where the movement of a single bead when the pump was driven is shown along the time axis. The bead progressed while oscillating during the same period as the pump.

### 3.2. Parallel Pump

The experimental results of the parallel pump are shown in Figure 7 and Figure 8, along with the results of the single pump for comparison. Figure 7 shows the movement of a single bead over time, whereas Figure 8 shows the distance covered by the beads moving forward and backward in the observation area under each driving condition. The length of the entire bar graph indicates the distance advanced, and the orange portion of the bar graph indicates the distance retreated. Data were collected for 20 beads and averaged. The backward distance decreased as the phase difference approached π, although the pulsation was greater than that of the single pump under all phase-difference conditions. When the phase difference was set to π, the pump was driven at a frequency of 20 Hz, whereas the beads pulsated at a frequency of 40 Hz.

### 3.3. Series Pump

The experimental results of the series pumps are shown in Figure 9 and Figure 10 along with the results of the single pump for comparison. Figure 9 shows the movement of a single bead over time, whereas Figure 10 shows the distance covered by the beads moving forward and backward in the observation area under each driving condition. Compared with the single-pump results, both phasing conditions reduced the rate of backward distance and pulsation.

## 4. Discussion

Table 1 shows the measured flow velocity of the beads and percentage of the backward distance relative to the forward distance for each experiment. Our proposed sidewall-driven micropump suppressed pulsation using a series configuration rather than a parallel configuration.

The parallel pumps showed a certain degree of backward motion under all phase-difference conditions. Overall, the closer the phase difference was to π, the smaller the backward motion was. This was likely because when the phase difference was large, the flow diverged, as shown in Figure 11, and the fluid sent from the other pump flowed into the pump in the negative pressure state. When comparing the parallel and series pumps, the series pumps were more effective in reducing pulsation. The other pump downstream of the series pump may have acted as a valve to prevent backflow, which is illustrated in Figure 12. When the movable wall of the upstream pump opened and drew in the surrounding fluid, the backflow was reduced because the downstream pump provided resistance. Pulsation may be reduced by installing a new valve near the pump outlet that can completely block the flow path. However, as explained in Section 2.1, it is difficult to completely block the flow path using the sidewall-driven method; therefore, two pumps are considered to be necessary if the flow path is created with a single layer. The series pumps exhibited small differences in each phase. This may have been because the effect of each pump acting as a valve was greater than the effect of the different phases of flow counteracting the reverse flow.

When the pump pressure was low, the wall pushed less fluid, which may have reduced the resultant distance traveled by the particles. Therefore, it may be possible to control the position of particles with even higher resolution while utilizing the same pump. However, lowering the pressure may have affected backflow because of the large throat in the flow path during pressurization. Therefore, a similar experiment was conducted using series pumps at a reduced applied pressure of 0.15 MPa. The results are shown in Figure 13. When the applied pressure was 0.20 MPa, the average distance moved by the beads in one cycle was 49.9 µm, whereas when the applied pressure was 0.15 MPa, the average distance was 19.5 µm. These results indicate that the speed of the particle and the distance traveled in a single step can be adjusted by changing the pressure. In addition, the magnitude of retraction varied depending on the phase-difference condition, with the smallest retraction occurring when the phase difference was close to zero. These results may be related to the fact that a lower applied pressure reduces the pump’s ability to hold the flow back. Considering that the phase-difference condition resulting in the smallest pulsation may vary depending on the distance between the two pumps, various flow paths should be created in the future to clarify which factors affect the size of the pulsation.

## 5. Conclusions

We attempted to reduce pulsation using two sidewall-driven micropumps in parallel and a series pump. The series pump reduced the pulsation compared to the single pump, and the optimum phase difference varied depending on the level of applied pressure. This proposed micropump can be fabricated in a single layer using simple soft lithography, making it easy to mass-produce. In the future, we will increase the number of experimental samples to determine the parameters affecting the phase-difference condition and further reduce pulsation. Thereafter, we will perform cell positioning experiments using a series pump rather than a parallel pump. To transport the target cell to the target position, the difference between the current position and target position will be obtained using image processing, and the applied pressure will be adjusted to the pump according to the difference using an electrically controlled pressure regulator. Using this image feedback method, we will conduct cell position manipulation experiments using a series pump and evaluate the performance of the pump.

## Figures and Tables

**Figure 1 micromachines-14-00019-f001:**
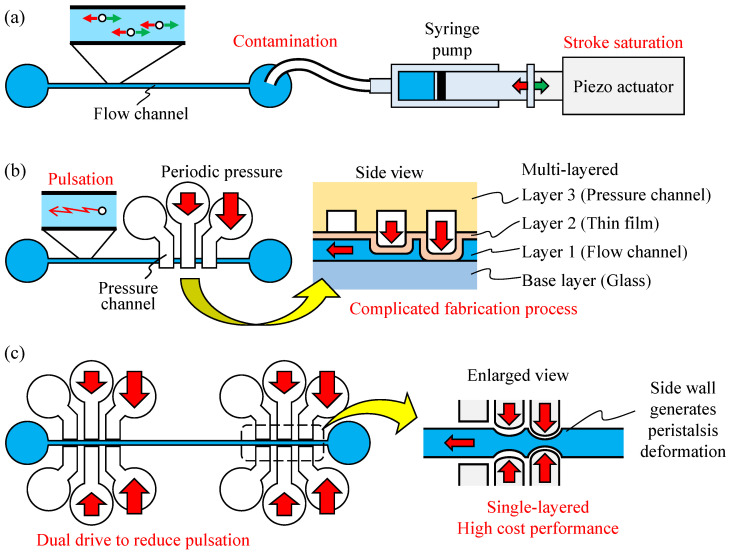
Particle position control systems, where (**a**–**c**) show a traditional system actuated by an external syringe pump, a multi-layered internal micropump, and a single-layer, dually driven micropump, respectively.

**Figure 2 micromachines-14-00019-f002:**
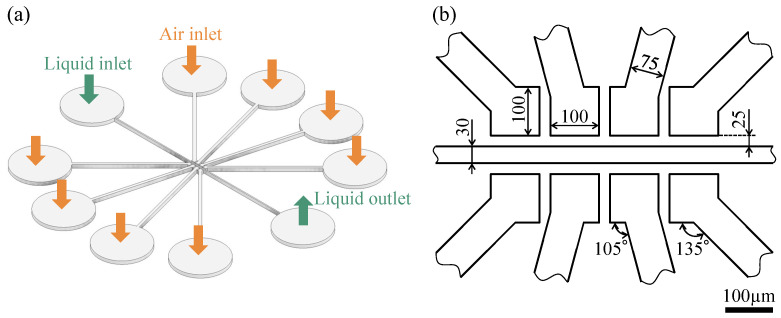
Sidewall-driven micropump: (**a**) overall view of the flow path; (**b**) dimensions of the sidewall-driven micropump.

**Figure 3 micromachines-14-00019-f003:**
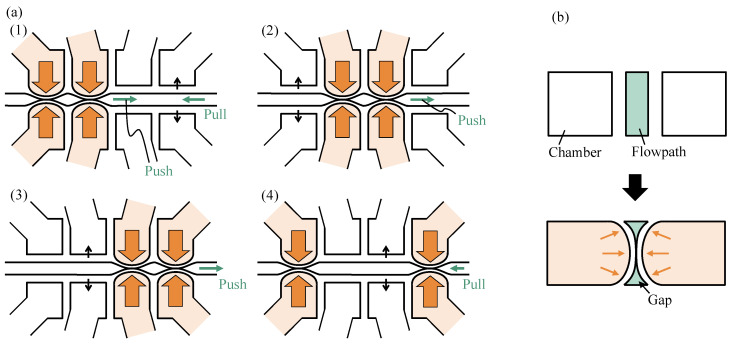
Relationship between peristaltic pump and pulsation: (**a**) pump pressurization sequence and pulsation; (**b**) cross-sectional view of the flow path.

**Figure 4 micromachines-14-00019-f004:**
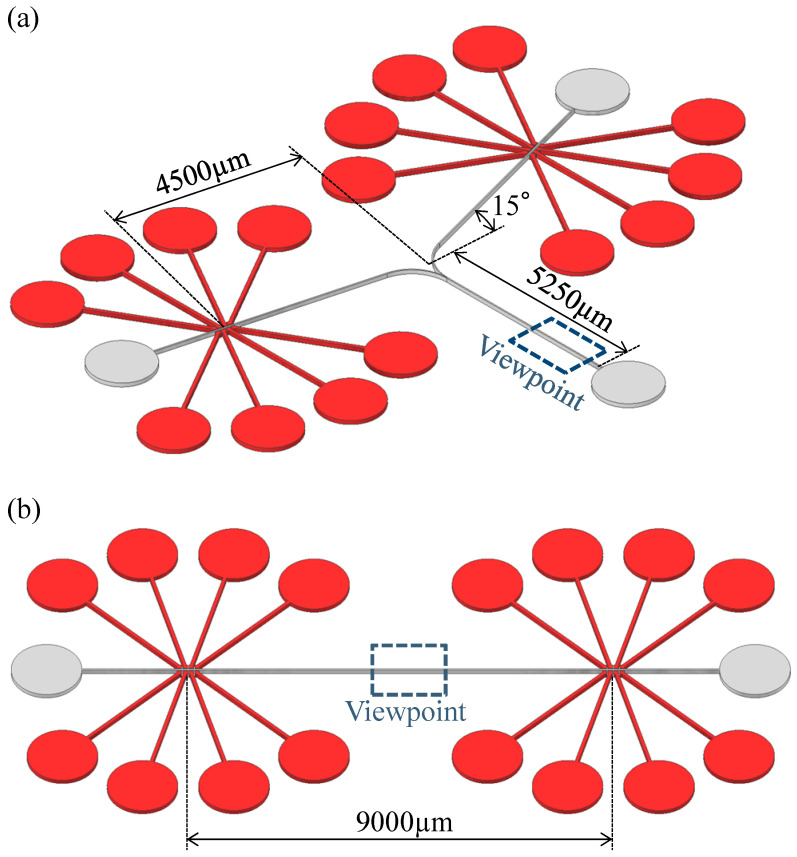
(**a**) Paralell pump. (**b**) Series pump.

**Figure 5 micromachines-14-00019-f005:**
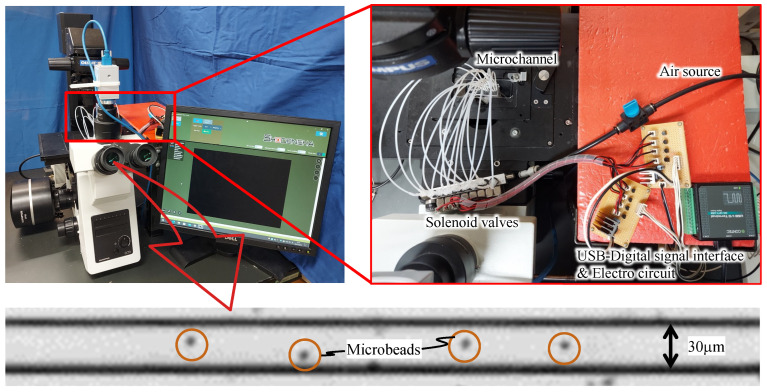
Experimental setup: overall experimental system and view from the microscope.

**Figure 6 micromachines-14-00019-f006:**
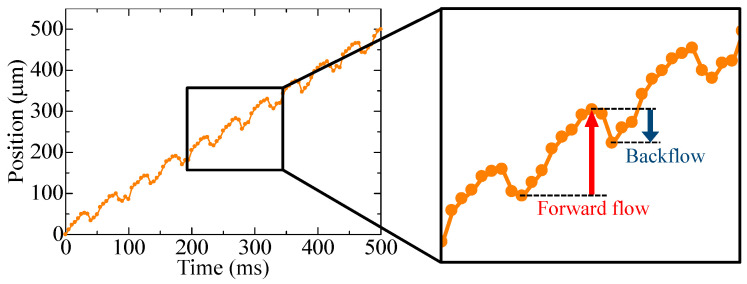
Bead position when driving a single pump, and forward flow and backflow in one period.

**Figure 7 micromachines-14-00019-f007:**
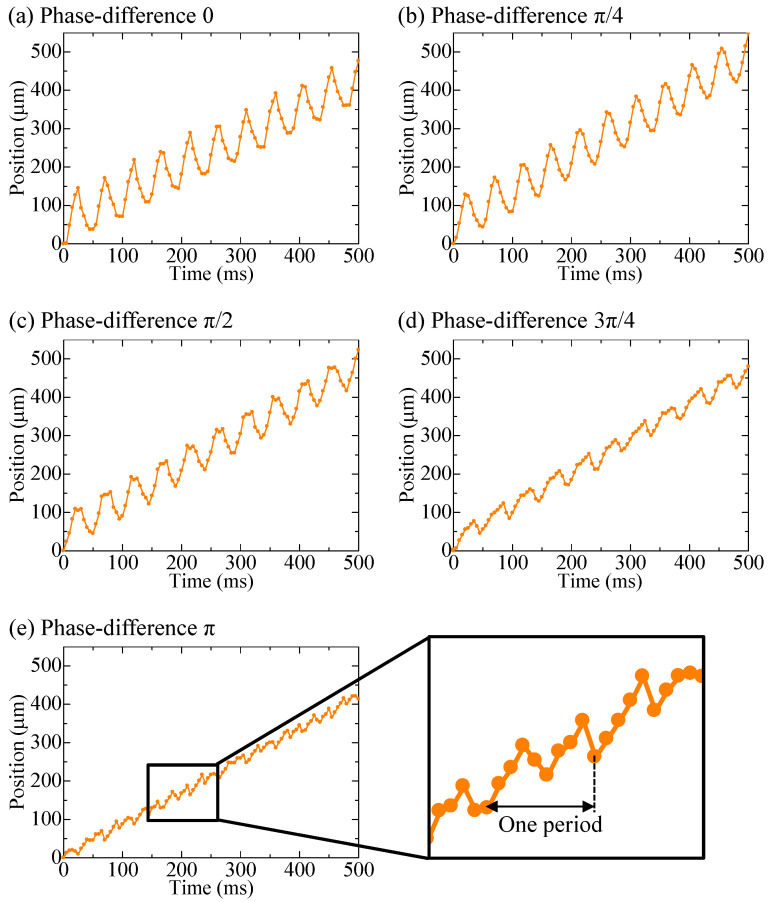
Bead positions when driving a parallel pump, where (**a**–**e**) show the phase differences of 0, π/4, π/2, 3π/4, and π, respecticely, and the enlarged graph shows one period at the phase difference of π, which was the condition for minimum pulsation.

**Figure 8 micromachines-14-00019-f008:**
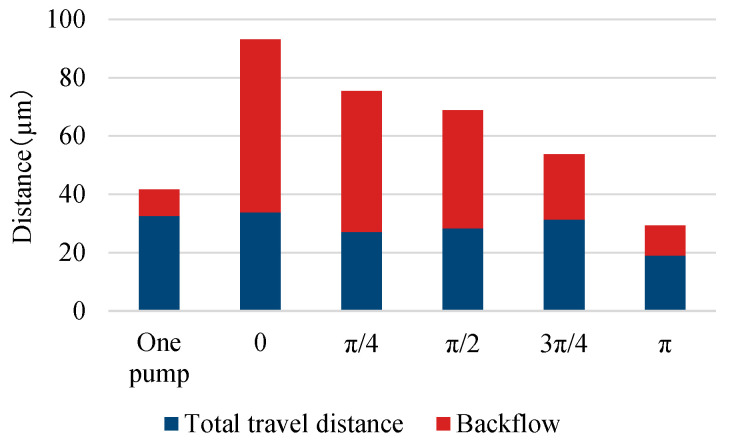
Forward flows and backflows of a parallel pump.

**Figure 9 micromachines-14-00019-f009:**
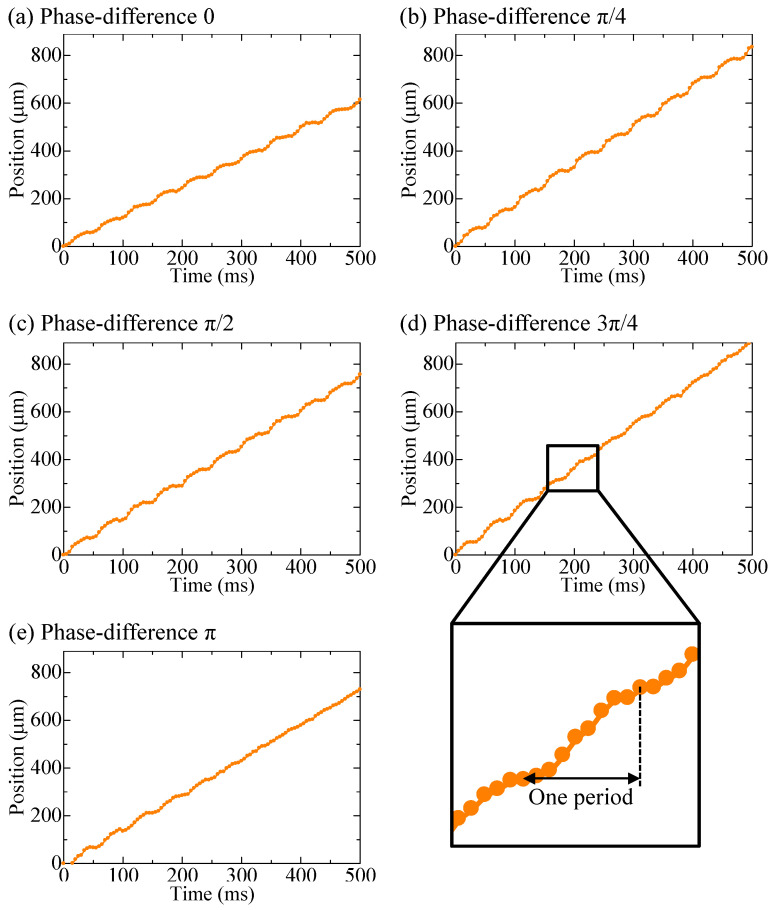
Bead positions when driving a series pump, where (**a**–**e**) show the phase differences of 0, π/4, π/2, 3π/4, and π, respectively, and the enlarged graph shows one period at the phase-difference of 3π/4, which was the condition for minimum pulsation.

**Figure 10 micromachines-14-00019-f010:**
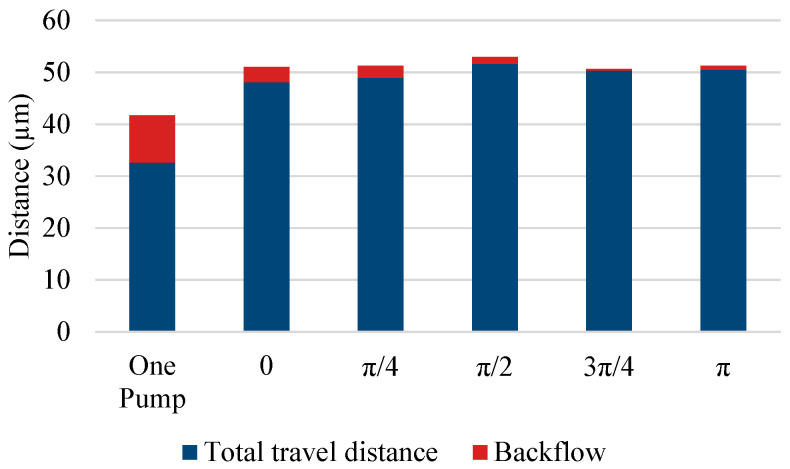
Forward flows and backflows of a series pump.

**Figure 11 micromachines-14-00019-f011:**
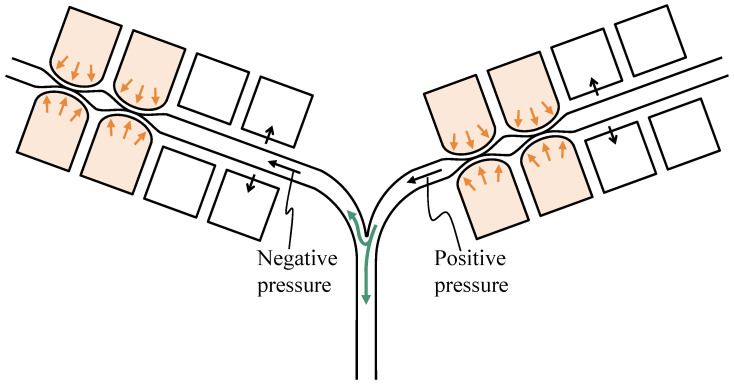
Mechanism for reducing backflow of the parallel pump.

**Figure 12 micromachines-14-00019-f012:**
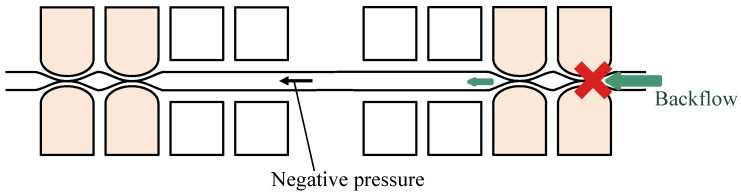
Mechanism for reducing backflow of the series pump.

**Figure 13 micromachines-14-00019-f013:**
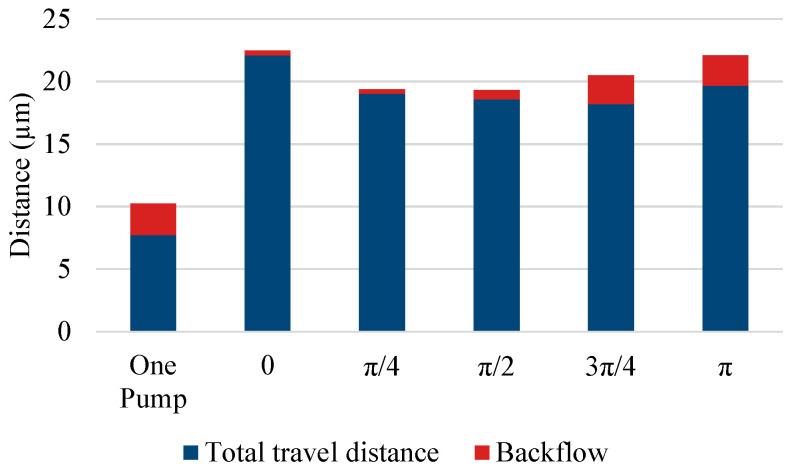
Experimental result of the series pump (pressure: 0.15 MPa).

**Table 1 micromachines-14-00019-t001:** Comparison of overall experimental results.

	Single	Parallel Pump					Series Pump				
	Pump	0	π/4	π/2	3π/4	π	0	π/4	π/2	3π/4	π
Flow rate (mm/s)	0.652	0.677	0.541	0.566	0.625	0.757	0.962	0.979	1.03	1.01	1.01
Percentage of backflow relative to forward flow	21.9%	63.7%	64.1%	58.9%	41.8%	35.3%	5.72%	4.41%	2.49%	0.597%	1.61%

## Data Availability

The datasets used and/or analyzed during the current study are available from the corresponding author upon reasonable request.

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
