# Peer review of "Pulsation Reduction Using Dual Sidewall-Driven Micropumps"

_micromachines, 2022, doi:10.3390/mi14010019_

Round 1

Reviewer 1 Report

The authors demonstarted an interesting approach to reduce the pulsation by utilising two sidewall-driven micropumps. I believe, this work is interesting and publishable after minor revision.

The abstract must be more clear. It should contain the content of the proper precisely.

The fonts in the figure is consistent.

Authors must put extra effort to rearrage and reorient the figures. A multiple figures can be merged into a large figure. Please reconsider the distribution of the figures as it will help the reader.

Author Response

We appreciate Reviewer #1 for taking time to review our paper and giving us valuable comments.

1. We added these explanations in the abstract. (In P.1 L.2-5 and L.13-14, respectively.)

"However, the current mainstream method using a syringe pump and piezoelectric actuator is not suitable for long-term experiments. There are also methods that incorporate a pump mechanism into a microfluidic channel, but they are not suitable for mass production due to their complex structure.”

“This will enable precise particle control and is expected to contribute to more inexpensive and easier cell manipulation experiments.”

2. We unified the font in Figure 1 with the others.

3. We merged the three figures in Figure 5 and merged the two figures in Figure 6.

We merged Figure 7e with the enlargement and added a pull-out line for clarity. We also changed caption of Figure 7.

We merged Figure 9d with the enlargement and added a pull-out line for clarity. We also changed caption of Figure 9.

We rotated Figure 11 by 90° for clarity.

Reviewer 2 Report

The authors proposed to reduce pulsation by using two sidewall-driven micropumps in parallel and a series pump. The series pump succeeded in reducing the pulsation compared to a single pump, while the results also indicated that the optimum phase difference varied depending on the level of applied pressure. This study is interesting and instructive to researchers who are keen on pulsation pumps. 

1. the authors should mention the non-mechanical pumps in their introduction. like some EHD pumps in the literature of fluidic rolling robots using voltage-driven oscillating liquid. 

2. the pump in their design is a peristaltic pump. Considering the pulsation, many companies already developed syringe pumps that have little pulsation. 

3. Their methods are interesting. In their methods, they used many pumps to reduce pulsation. In my opinion, with only a small valve placed on the outlet of the single pump, the pulsation will be reduced since the peristaltic pump consists of three on-chip valves. 

4. the results in table.1 are interesting. The serialized pumps are beneficial for reducing pulsation. I think they should mention some controlling strategies in their paper, especially if they are using parallelized and serialized pumps. 

Author Response

We appreciate Reviewer #2, for valuable and helpful comments that significantly improved our paper.

We used an English editing service to make significant corrections to our English. A certificate of English editing is attached.

1. We added this explanation in P.2 L.55-61 and citations #17-22.

“Another example of a non-mechanical pump is the electrohydrodynamic (EHD) pump. This method uses electrohydrodynamic phenomena, which are created by applying a voltage to the flow path, to move the fluid. Unlike the electroosmotic pumps described above, dielectric fluids are generally used as the working fluid. EHD pumps are used in various applications such as micromixers and droplet generation. However, as with electroosmosis pumps, the separation of the pump from the cell and the complexity of the structure are problems for cell manipulation.”

2. We added this explanation in P.2 L.38-45 and citation #9.

“Many companies offer external pumps that can be applied to microfluidic channels as an alternative to conventional methods. For example, a microfluidic pump developed by Blacktrace is capable of pulseless pumping and can be used for applications such as droplet generation. However, these commercially available pumps are designed for continuous flow and are not suitable for applications that require fast response, such as position control. In addition, contamination must be avoided when conducting cell experiments. To avoid mixing the liquid in the pump with the liquid in the channel by connecting the pump directly to the channel, the pump must be integrated into the channel device.”

3. We added this explanation in P.10 L.207-211.

“Pulsation could be reduced by installing a new valve near the pump outlet that can completely block the flow path. However, as explained in Chapter 2, Section 1, it is difficult to completely block the flow path with the sidewall-driven method, so two pumps are considered necessary if the flow path is created with a single layer.”

4. We added this explanation in P.11 L.236-244.

“In the future, we will increase the number of experimental samples to determine the parameters that affect the phase difference condition and further reduce pulsation. Thereafter, we aim to achieve cell positioning experiments using a series pump instead of a parallel pump. To transport the target cell to the target position, the difference between the current position and the target position is obtained using image processing, and the applied pressure is adjusted to the pump according to the difference using an electrically controlled pressure regulator. Using this image feedback method, we will conduct cell position manipulation experiments using a series pump and evaluate the performance of the pump.”

Round 2

Reviewer 2 Report

The authors addressed my questions carefully.